# Challenges of Explaining Control

**Adrian Agogino[1], Ritchie Lee[2], Dimitra Giannakopoulou[1]**

[1]NASA Ames Research Center
[2]Stinger Ghaffarian Technologies, Inc. (SGT)
NASA Ames Research Center, MS 269-1
Moffett Field, California 94035
{adrian.k.agogino,ritchie.lee,dimitra.giannakopoulou}@nasa.gov

## Abstract

Reinforcement learning and evolutionary algorithms are used increasingly in the development of sophisticated control solutions for autonomous systems. However, it is challenging to trust such solutions for safety-critical systems because the rationale behind the control decisions they produce is obfuscated, and hidden behind parameters that are not directly related to the problem they target. Several approaches have been proposed to explain standard supervised learning algorithms, but these approaches cannot be readily applied to control algorithms due to the time-extended nature of the latter. This paper experiments with six techniques in order to develop explanations for autonomous, learning-based control: 1) Bayesian rule lists, 2) Function analysis, 3) Single time step integrated gradients, 4) Grammar-based decision trees, 5) Sensitivity analysis combined with temporal modeling with LSTMs, and 6) Explanation templates. These techniques are tested on a simple 2d domain, where a simulated rover attempts to navigate through obstacles to reach a goal. For control, this rover uses an evolved multi-layer perception that maps an 8d field of obstacle and goal sensors to an action determining where it should go in the next time step. Results show that some simple insights in explaining the neural network are possible, but that good intuitive explanations are difficult.

## Introduction

Explanation of machine learning algorithms is a challenging and important field of research. Most techniques to date have focused on supervised learning algorithms, such as image processing, text processing and medical diagnosis (Letham et al. 2015; Gunning ). Instead of supervised learning, this paper focuses on reward based machine learning such as reinforcement learning and evolutionary algorithms, where rewards are given to measure performance instead of using examples of what is correct. The nature of reward learning and supervised learning is different in both problem domains and learning tools used to solve these problems. In this paper we look at explainability techniques that have been designed for supervised learning problems and apply them to reward learning problems.

Reinforcement learning and evolutionary algorithms can be used to automatically learn high performance control systems for complex problems (Floreano and Mondada 1994; Crites and Barto 1996; Agogino, Stanley, and Miikkulainen 2000). This is particularly the case in the context of autonomy where control may involve many variables and need to dynamically adapt to different environments and situations.

A common form of machine learning is to train a set of weights of a neural network-based control policy. Based on inputs (such as sensors) the control policy can command control actions (such as speed and direction of a vehicle). Training is typically done with a simulator, where the learning algorithm attempts to improve the performance of the control policy through a long series of trials. The goal of this training process is to produce a high-performance non-linear control policy that takes inputs and produces controls.

While a successful training will produce a control policy that achieves high performance in simulation, how the control policy actually works will typically be unclear to its programmers, let alone its end-users. Due to this fact, machine learning algorithms are often referred to as "blackbox": their inputs and outputs can be viewed, but there is no knowledge of their internal workings.

Even when machine learning achieves high performance, it can be difficult to trust for two reasons: 1) coverage, and 2) generalizability. In terms of coverage, while an algorithm may have performed well in scenarios that were tested, there may be other likely scenarios where it would have performed very poorly. In addition since coverage of machine learning algorithms is largely dependent on the data set, the user may not even be aware of the algorithm's coverage and can easily overlook large gaps in the data sets. In terms of generalizability, while the algorithm performed well in the simulator it may not perform well in the real world or in environments that are slightly different than the simulated one. These problems can be exacerbated by the blackbox nature of these learning algorithms, where reward hacking, poorly defined utility functions or simple errors in the simulator can lead to unrealistically high levels of performance that cannot be achieved when deployed. In addition, machine learning algorithms have many unintuitive parameters that have no obvious relation to the underlying control problem, such as

number of hidden nodes and learning rates. Yet poor choices of these parameters can lead to poor generalization.

Improving explainability of these blackbox algorithms can help improve trust that they will behave as expected when deployed (Gunning ). If a control decision is backed up by a meaningful and understandable rationale, then one can trust that the decision is not made "by chance", and therefore the system can be expected to behave well in other similar circumstances. Additionally, if we understand a learned control algorithm, we can see if there are any clear gaps in coverage, or if there are any obvious flaws that would prevent it from generalizing outside of the simulated environment. On the other hand, what constitutes a meaningful, understandable explanation?

Providing explanations of machine learning is a very active research field. Several approaches have been proposed for standard supervised learning algorithms. Despite this fact, it is still unclear what types of explanations may be suitable in practice. Control further complicates the picture, because control strategies develop over time, and are typically not evaluated over snapshots. How can such strategies be captured in explanations and what type of explanations would those be?

To address this problem, we have experimented with a variety of techniques to provide explanations in the context of a very simple machine learning algorithm that we developed for navigating a rover towards a goal while avoiding obstacles. We decided to build the algorithm from scratch in order to evaluate the pitfalls and errors that may occur in developing such systems, as well as how/what explanations may assist in detecting those. We used six techniques in order to develop explanations: 1) Bayesian rule lists, 2) Function analysis, 3) Single time step integrated gradients, 4) Grammar-based decision trees, 5) Sensitivity analysis combined with temporal modeling with LSTMs, and 6) Explanation templates. This set of techniques was chosen as it represents a diverse set of explanations that could be readily applied to control data. In particular, it includes both local and global explanations. These local attempt to explain a single control action in a particular state. To form a big picture of a control policy with local explanations, we would want many local explanations covering many different states. In contrast global explanations try to explain an overall action policy over all states.

The remainder of the paper is organized as follows. We first present the example obstacle avoidance problem we use throughout the paper. Then we describe the neural network controller and the Monte Carlo algorithm used to determine the weights of the neural network. We subsequently discuss the need for explainability and how simple analysis of algorithm performance may be insufficient. To address this we present six different explainability algorithms applied to the example problem and discuss their relative merits.

## Test Problem

We test our explainability methods on a simple test problem, where a rover moving on a 2d plane tries to navigate towards a goal while avoiding obstacles. It does this with a neural network that maps goal and obstacle sensors into a control action that determines the speed and direction of the rover for the next time step. The weights of this neural network are determined with an evolutionary algorithm using a simulation of the environment.

## Environment and Utility

In our test domain, a rover attempts to reach a single goal while avoiding 100 obstacles placed randomly on an x-y plane (see Figure 1). The rover starts in the middle of the obstacle field and the goal is located above the obstacle field. At each time step the rover takes a small movement in the x and y direction. At the end of 70 time steps, the rover's performance is evaluated.

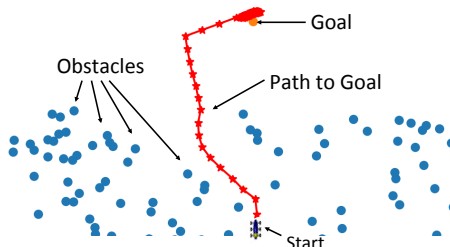

Figure 1: Obstacle Avoidance Problem. A rover attempts to navigate towards a goal while avoiding obstacles in a 2-d plane.

## Sensors

At every time step, the rover senses the world through eight continuous sensors (Agogino and Tumer 2004). From a rover's point of view, the world is divided up into four quadrants with fixed orientation to the x-y axis, with two sensors per quadrant (see Figure 2 Left). For each quadrant, the first sensor returns a function of the obstacles in the quadrant at time $t$. Specifically the first sensor for quadrant $q$ returns the sum of inverse square distances from an obstacle to the rover:

$$s_{1,t} = \sum_{j \in J_q} \frac{1}{\delta_j{}^2} \ ,$$

where $J_q$ is the set of obstacles in quadrant $q$ and $\delta_j$ is the euclidean distance from obstacle $j$ to the rover. The second sensor, $s_{2,t}$, returns the inverse square distance from the rover to all the goals in each quadrant at time $t$. In our case since there is only one goal, only the quadrant that contains the goal will have a non-zero value, which is $1/d^2$ where $d$ is the euclidean distance from the rover to the goal.

The sensor space is broken down into four regions to facilitate the input-output mapping. There is a trade-off between the granularity of the regions and the dimensionality of the input space. In some domains the tradeoffs may be such that it is preferable to have more or fewer than four sensor regions.

## Rover Control Strategies

With four quadrants and two sensors per quadrant, there are a total of eight continuous inputs. This eight dimensional

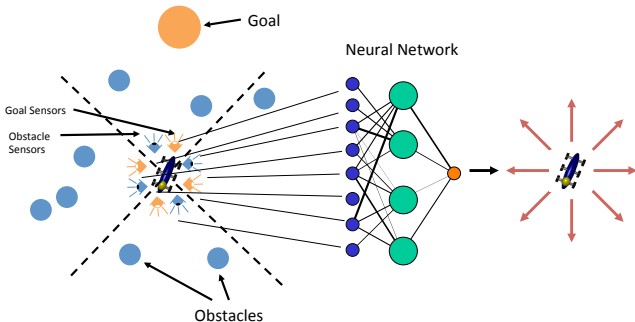

Figure 2: Rover Sensors. The rover has 8 sensors: 4 obstacle sensors and 4 goal sensors. Each sensor observes the presence of objects in its quadrant based on a sum of inverse squared distances. More objects and closer objects there increase sensor value. Sensors are inputs to a neural network that produces a control action determining x, y movement in the next time step.

sensor vector constitutes the state space for a rover. At each time step the rover uses its state to compute a two dimensional output. This output represents the $x, y$ movement relative to the rover's location and orientation.

The mapping from rover state to rover output is done through a Multi Layer Perceptron (MLP) (Haykin 1998), with eight input units, ten hidden units and two output units [1]. The MLP uses a sigmoid activation function, therefore the outputs are limited to the range $(0, 1)$. The actual rover motions $dx$ and $dy$, are determined by normalizing and scaling the MLP output by the maximum distance the rover can move in one time step. More precisely, we have:

$$dx = 2d_{max}(o_1 - 0.5)$$
$$dy = 2d_{max}(o_2 - 0.5) \ ,$$

where $d_{max}$ is the maximum distance the rover can move in one time step, $o_1$ is the value of the first output unit, and $o_2$ is the value of the second output unit.

### Monte Carlo Algorithm

We use a simple 2-phase Monte Carlo algorithm to determine the weights for the neural network controller. In the first phase, the weights for each Monte Carlo run are set to a value between -6 and 6 sampled from a uniform distribution. After the weights are selected the rover is evaluated in a simulation for 70 time steps, and its performance is recorded. 1000 Monte Carlo runs are performed in this first phase.

In the second phase, for each Monte Carlo run, the weights are copied from the neural network with the best performance in the first phase. The weights of this copy are then mutated by adding noise selected from a uniform distribution in the range -0.05 to 0.05. 200 Monte Carlo runs are performed from this second phase and the weights of the

---

[1]Note that other forms of continuous reinforcement learners could also be used instead of evolutionary neural networks. However neural networks are ideal for this domain given the continuous inputs and bounded continuous outputs.

best performing rover are saved as the final solution of the algorithm.

The reward used to evaluate each Monte Carlo run is as follows:

$$R = \sum_{t=0}^{T} 100s_{2,t} - s_{1,t}$$

where $s_{2,t}$ is the goal sensor value and $s_{1,t}$ is the sum of obstacle sensor values at time $t$ run for $T$ time steps. This utility goes up when the rover gets closer to the goal and down when it gets close to obstacles. Note that the goal sensor is scaled since the rover is usually much closer to obstacles than the goal.

### Explainability

At the completion of the Monte Carlo algorithm we have a neural network capable of controlling a rover in our test problem. Traditionally we would test this controller by running it and observing how it performs, such as by looking at the path it took as shown in Figure 1. We can also look at how its performance improved during training. For instance Figure 3 shows that while random neural networks tend to perform poorly, there are a few that perform much better than average. The figure also shows that in Phase 2 of training that performance improves, but not significantly.

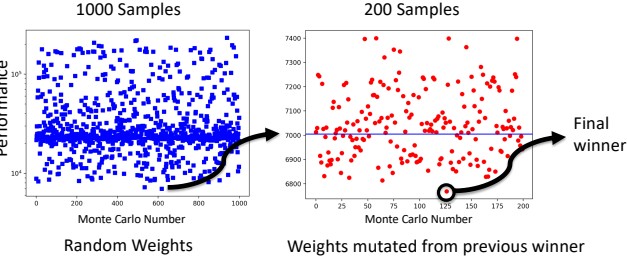

Figure 3: Rover Performance during training, displayed as negative reward. Left: In the first 1000 Monte Carlo runs, performance varies considerably. Right: The next 200 Monte Carlo runs are based on the best sample from the first thousand (performance of this sample shown in blue line).

While this analysis gives some insight into the performance of the neural network controller, it does not tell us how it actually operates. In particular, it does not tell us if the controller has any hidden failure modes that we should be aware of. Machine learning algorithms and neural networks in particular can have many subtle failures that not be apparent in basic testing. A neural network is represented by a large collection of interconnected weights and inspecting the values of these weights is usually not helpful in determining if the network is operating correctly. As an example of such a failure, when we first trained our neural network on the obstacle avoidance problem, we accidentally limited the range of possible weight values to be on too narrow of interval for the neural network to fully approximate non-linear functions. While the training went smoothly and the algorithm produced a viable control policy, the performance of this control policy was significantly lower than what it could

have been due to this error. Ultimately a unit test, testing the ability of the neural network to approximate a sine wave revealed this issue.

While performance tests and unit tests give some insight into how a neural network is operating, we would like additional explanations of how a trained neural network is actually operating. In this paper we attempt to analyze our neural network using several different explainability methods: 1) Bayesian rule lists, 2) Function analysis, 3) Single time step integrated gradients, 4) Grammar-based decision trees, 5) Sensitivity analysis combined with temporal modeling with LSTMs, and 6) Explanation templates. This set of techniques was chosen as it represents a diverse set of explanations that could be readily applied to control data. In this set, integrated gradients provides a local explanation that attempts to explain a single control action in a particular state. The rest of the explanations are more global in that they attempt to explain the overall control policy independent of state. Another factor is temporal as a control policy attempts to maximize reward over time. Of our explanations only grammar-based decision trees and modeling with LSTMs explicitly attempts to reason over time. In general, the temporal aspect of control makes explanations difficult and complex, therefore most of our explanations attempt to explain individual actions rather than an entire sequence of actions.

## Bayesian Rule Lists

Explanations in terms of Bayesian Rule Lists (BRL) (Letham et al. 2015) consist of a list of if-then rules predicated on the controller's inputs. These rules are generated looking at the input/output data associated with the controller, not looking at the neural network itself. Since the domain is continuous and the rules are discrete, a mapping between the domain and the rules needs to be created.

For our example problem we created a simple mapping by hand. For the obstacle sensors, we converted the values of the four quadrants into one of four categorical labels (up, down, left and right), signifying which quadrant had the greatest value. For instance if left quadrant had the greatest value then the label would be "left." The outputs of the controller are converted to two binary values corresponding the x and y values of the output. When the y output has a positive value then its label is 1, otherwise its value is 0. The x value is encoded similarly. To simplify the mapping we ignore the values of the goal sensor and tested the rover in an area where the obstacle sensors dominated. Given these mappings we can convert a set of sensor and control action data into a set of labels.

We performed a BRL extraction using a control run of 70 time steps. Doing this, we can generate four separate rules for going up, down, left and right. The results for the up rule were as follows:

```
if obstacles to left
    go up with probability .19
else if obstacles are up
    go up with probability .87
else
    go up with probability .07
```

Notice that the second rule is somewhat problematic. The neural network actually wants to head towards an obstacle when it is close by. On further inspection, we saw that indeed the rover tends to head towards obstacles, but also turns enough as it is doing so to avoid the obstacle. While effective, this strategy would not seem satisfactory for safety critical systems.

While the BRL was able to expose a potential hazard in the controller, it tended to be hard to use and did not give much insight into the full behavior of the controller. In addition, since it treats the controller as a black box, BRL is only able to characterize observed behavior and could miss important properties of the controller that were not observed in the training data.

## Activation Analysis

Our next analysis of the neural network controller is to look at the shape of the input/output functions. For each quadrant in the sensor field there are three functions. The up sensor functions are as follows (the other mappings are appropriately rotated according to the sensor orientation) 1) Mapping from obstacle sensor to Y control action, 2) Mapping from obstacle sensor to X control action, and 3) Mapping from goal sensor to Y control action. The plots of these functions are shown in Figure 4. From the plot we can see that the goal sensor controller behaves as expected. When the goal is present in a sensor, the controller tends to move towards the goal. However the obstacle sensor controllers are a bit more non-intuitive. When the rover gets close to obstacles in the up direction, the X controller will move the rover to the right. However if it gets very close to the obstacles the X controller will start moving in the left direction. Even more worrisome is that when obstacles are close the Y controller will accelerate towards them. This analysis confirms the explanation rules created by the Bayesian Rule List.

## Sensitivity Analysis

One way to test some of the properties of a neural network directly is to test the sensitivity of the inputs to the outputs (Tulio Ribeiro, Singh, and Guestrin 2016; Sundararajan, Taly, and Yan 2017). This analysis may be able to tell us for a particular location, which inputs are the most important to the controller's decision.

**Gradient Analysis** The most basic form of sensitivity analysis is gradient analysis where we measure the gradient of the input with respect to the outputs. This can be accomplished in neural networks using backpropagation. To test this analysis we created a scenario where a rover is located right below the location of an obstacle (see Figure 5). In this scenario we then calculated the gradient of each of the four obstacle sensors with respect to the controller output. The results are as follows:

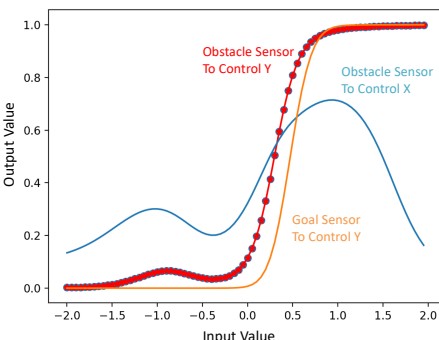

Figure 4: Neural Network Controller Functions. Function analysis shows rover should move towards goal as expected. However, rover also has a tendency to move towards obstacles. It only avoids them by turning when it gets close.

```
Up:             0.031
Left:           0.227
Down:           0.120
Right:          0.139
```

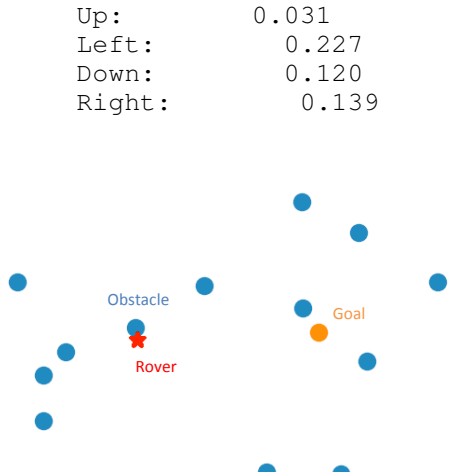

Figure 5: Scenario for Sensitivity Analysis.

This result shows the limitation of basic gradient analysis. We would expect the Up sensor to be the most important to the control, since there is an obstacle very close in the up direction. However, since the rover is so close to the obstacle this sensor saturated so any small change in its value causes almost no change control action. Therefore this sensor actually has the smallest gradient, which is the opposite of what we would hope in terms of explainability. Instead of looking at only local changes in its value we need to look at the effects of larger changes.

**Integrated Gradients** Integrated gradients attempts to solve this limitation of local gradients by adding a series of gradients from the sensor value of interest down to a baseline sensor value. In this way any important change that happens along this path will be recorded. To test integrated gradients we perform a test where the baseline sensor has a value of zero in all four quadrants and calculate 100 gradients from rover position in our scenario down to the baseline value. The results are as follows:

```
Up:             14.67
Left:           16.40
Down:           11.10
Right:          12.00
```

These results are somewhat more satisfying as the up sensor now has the second largest value.

**Explanation Template**

Our next attempt at explaining the behavior of the neural network is to model its global properties with respect to an understandable control algorithm (Chandrasekaran, Tanner, and Josephson 1989). We call this control algorithm an "explanation template." This template comprises a simple control algorithm that is easy to comprehend with free parameters that are determined by analyzing the behavior of the neural network. We tried this technique using a simple linear policy. Here is the policy template for the upward looking goal and obstacle sensors:

$$v_{up} = w_0 s_{o,u} + w_1 s_{g,u}$$
$$v_{right} = w_2 s_{o,u}$$

where $v_{up}$ and $v_{right}$ are the up and right velocities for the next time step, $s_{o,u}$ is the value for the upward looking obstacle sensor, $s_{g,u}$ is the value for the upward looking goal sensor, and $w_0, w_1, w_2$ are the free parameters. Using data from 50 trials of the rover we performed linear regression and found the values of the free parameters producing the following explanation of the system:

$$v_{up} = \frac{s_{o,u}}{803} + \frac{s_{g,u}}{6139}$$
$$v_{right} = \frac{s_{o,u}}{1585}$$

This explanation shows that the neural network has a small tendency to move towards the goal, but a large tendency to move towards an obstacle. It is able to avoid obstacles as it also has a tendency to move right when it approaches an obstacle. These findings are consistent with the function analysis and the Bayesian rule lists.

**Grammar-Based Decision Trees**

Our next attempts use grammar-based decision trees (GB-DTs) (Lee et al. 2018). The idea is to learn an interpretable model from data and then inspect the learned rules to gain insight into system behavior. GBDT generalizes a traditional decision tree, where the decision rules are Boolean expressions derived from a context-free grammar. The grammar allows any logical language to be used and the user can tune the grammar for explainability. GBDT has been shown to provide good representational ability while being interpretable (Lee et al. 2018). GBDT can model different types of data by choosing an appropriate grammar. For example, a grammar based on first-order logic can be used to model static data, while a grammar based on temporal logic can be used to model time series data. We explored two approaches to applying the GBDT model. The first approach models the input-output behavior of the neural network policy and the second approach models the time series data that the policy and its environment together produces.

**GBDT Control Policy Modeling**  In this first GBDT approach, we model the input-output behavior of the neural network policy. We learn an interpretable model that approximates the behavior of the policy and then inspect the learned rules to gain insight into the decisions of the policy.

To construct the training data for the GBDT, we use the input-output pairs of the neural network policy seen during its training. Since the output of the neural network policy is a relative position in 2d, but GBDT can only produce discrete output, we take the relative angle of the network output and round it to the nearest 45 degrees. We use a simple grammar consisting of comparison operators less than $<$ and greater than $>$ operating on the input features $xid$; and logical operators conjunction $\wedge$, disjunction $\vee$, and negation $\neg$ that enable the formation of more complex expressions. The full grammar is shown in Figure 6.

$$b \mapsto (b \wedge b) \mid (b \vee b) \mid \neg b$$
$$b \mapsto (X[xid] < X[xid]) \mid (X[xid] > X[xid])$$
$$xid \mapsto top \mid left \mid bottom \mid right$$

Figure 6: GBDT Grammar for Modeling Control Policy.

The GBDT was trained using genetic programming (Koza 1992) to optimize each rule of the tree (Lee et al. 2018). The resulting GBDT, shown in Figure 7, attained 85.5% accuracy. The GBDT found two rules to distinguish between three policy outputs up, up_left, and up_right. The reason there are only three actions used is because the goal is located above the start point, so those are the primary actions required for successful navigation. In Figure 7, if there are more obstacles to the left than to the bottom, then go up and to the left. This behavior can be observed in the two up_left segments in Figure 1 as the agent navigates toward and around the cluster of the obstacles to the left. The second rule has two terms. If bottom is greater than right and top is greater than left, then move up. This behavior is seen in Figure 1 as the rover moves away from obstacles to the bottom passing obstacles to the right. As the rover approaches the goal, there are no obstacles to the top or left, so top equals left and the second rule becomes false. In this case, the output is up_right.

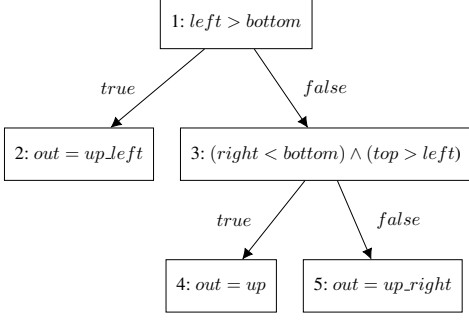

Figure 7: GBDT Result from Modeling Control Policy.

Our result reveals that the learned neural network policy may be overfitted to the scenario because the output relies on this specific arrangement of the obstacles. The discovered rules are indeed true patterns in the data. However, it is unclear that these rules provide satisfactory explanations to humans. For example, humans do not find comparisons between different axes, such as left $>$ bottom, very intuitive.

**GBDT Temporal Modeling**  The above approach does not take into account (1) the temporal nature of the problem and (2) the interactions between the controller and the environment. In this second GBDT approach, we attempt to capture the temporal properties of the combined controller-environment system. We construct a training dataset where the inputs are multivariate sequences of the obstacle sensor values, goal sensor values, policy output, and agent position, and the target outputs are whether the sequence was produced by the final (optimal) neural network control policy or another (suboptimal) controller that was considered but ultimately discarded during training. We train a GBDT model on the temporal data and then inspect the learned rules to gain insight into the temporal properties that distinguish between paths from the optimal and suboptimal controller.

We specify a grammar based on a simple temporal logic as shown in Figure 8. The grammar includes temporal operators globally $G$ and eventually $F$; elementwise logical operators conjunction $\wedge$, disjunction $\vee$, negation $\neg$; and comparison functions that perform elementwise comparison of a feature sequence to precomputed constants. These comparison functions are expressed in the grammar in the form $f_{op}(xid, vid)$, which computes $X[xid] \, op \, V[xid, vid]$, where $X[xid]$ is the temporal sequence of feature $xid$, $op$ is a comparison operator, and $V[xid, vid]$ is a precomputed lookup table that returns the $vid$'th decile division point of the range of feature $xid$ in the data.

The GBDT was trained using genetic programming (Koza 1992) to optimize each rule of the tree (Lee et al. 2018). The resulting GBDT, which attained 99.9% accuracy, is shown in Figure 9. The GBDT model identified three temporal properties relevant to distinguishing between whether a sequence is optimal or suboptimal. The following properties need to be simultaneously satisfied for the input sequence to be classified as optimal: (1) At some point, $action\_x$ reaches a value that is greater than 90% of the range of $action\_x$ in the data (node 1 in Figure 9); (2) The following statement must be false: At some point, $obs\_sense\_right$ is greater than 30% of the range of $obs\_sense\_right$ in the data (node 2)(In other words, $obs\_sense\_right$ must be globally below 30% of its range); and (3) At some point, $action\_y$ is greater than or equal to 80% of the range of $action\_y$ in the data (node 4).

In summary, the GBDT has discovered that strong right and strong up actions combined with a weak right obstacle sensor are correlated with the optimal policy. While these properties hold true in the data, it did not provide a very deep insight or satisfying explanation for the control policy.

$$b \mapsto G(\vec{b}) \mid F(\vec{b})$$
$$\vec{b} \mapsto (\vec{b} \wedge \vec{b}) \mid (\vec{b} \vee \vec{b}) \mid \neg \vec{b}$$
$$\vec{b} \mapsto (\vec{r} < \vec{r}) \mid (\vec{r} \leq \vec{r}) \mid (\vec{r} > \vec{r}) \mid (\vec{r} \geq \vec{r})$$
$$\vec{b} \mapsto f_<(xid, vid) \mid f_\leq(xid, vid)$$
$$\vec{b} \mapsto f_>(xid, vid) \mid f_\geq(xid, vid)$$
$$\vec{r} \mapsto X[xid\_sens] \mid X[xid\_pos]$$
$$xid \mapsto xid\_sens \mid xid\_pos$$
$$xid\_sens \mapsto xid\_obs \mid xid\_goal$$
$$xid\_obs \mapsto obs\_sens\_top \mid obs\_sens\_left$$
$$xid\_obs \mapsto obs\_sens\_bottom \mid obs\_sens\_right$$
$$xid\_goal \mapsto goal\_sens\_top \mid goal\_sens\_left$$
$$xid\_goal \mapsto goal\_sens\_bottom \mid goal\_sens\_right$$
$$xid\_pos \mapsto xid\_action \mid xid\_loc$$
$$xid\_action \mapsto action\_x \mid action\_y$$
$$xid\_loc \mapsto x \mid y$$
$$vid \mapsto \mid (1 : 10)$$

Figure 8: GBDT Grammar for Temporal Modeling.

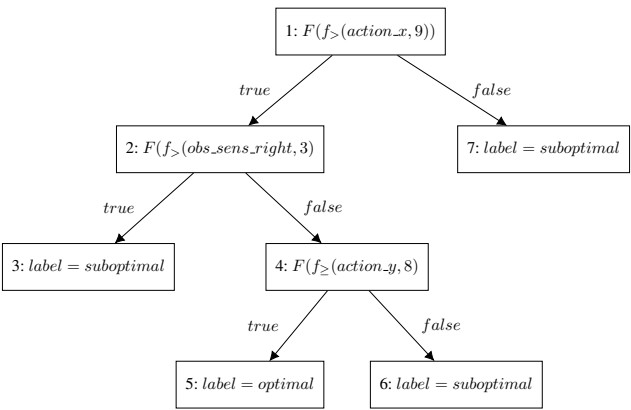

Figure 9: GBDT Result from Temporal Modeling.

## Temporal Modeling using LSTMs

This approach combines temporal modeling with attributions to highlight the most salient sequential inputs. We begin by training a long short-term memory (LSTM) classifier (Hochreiter and Schmidhuber 1997) to distinguish between input sequences produced by the final (optimal) neural network controller and sequences produced by another (suboptimal) controller considered (but ultimately discarded) during training. Then we apply integrated gradients (Sundararajan, Taly, and Yan 2017) to evaluate the importance of each input. Because LSTM is a neural network for sequential data, attributions highlight not only which features are important, but also at which time steps. Attributions produce local explanations in that explanations apply to a particular example, rather than explaining global patterns over the

dataset.

Figure 10 shows an interesting attributions result that occurs in many examples classified as optimal in the data. In this example, we see a clear repeating pattern in features 7 and 8 that is highlighted by attributions as being the most important in classifying this example as being produced by the optimal controller. Feature 7 is the bottom goal sensor and feature 8 is the right goal sensor. It is also observed that the attribution assigns more importance to the latter parts of the sequence. We investigated the highlighted values in the data and discovered that there is an interesting phenomenon in the data. Sequences produced by the optimal controller reaches the goal much sooner than 70 time steps. To collect maximum reward, the agent stays near the goal for as long as possible. However, because a complete stop is difficult to learn, the controller learned to cycle near the goal, and it is this cycling that is being highlighted by the attributions. The controller has learned the following behavior: (1) When the goal is near and located below, move a small amount downwards and to the left; and (2) when the goal moves from being detected by the bottom sensor to being detected by the right sensor, then jump up and rightward and restart the cycle. Indeed, the optimal controller exhibits this cycling behavior while the suboptimal ones do not.

While this approach using temporal modeling has demonstrated that it can help identify interesting patterns in the data, the algorithm merely highlights parts of the data and does not elaborate on why those parts are important. Ultimately, a human must perform additional analysis to try to understand the relevance, which may be very challenging.

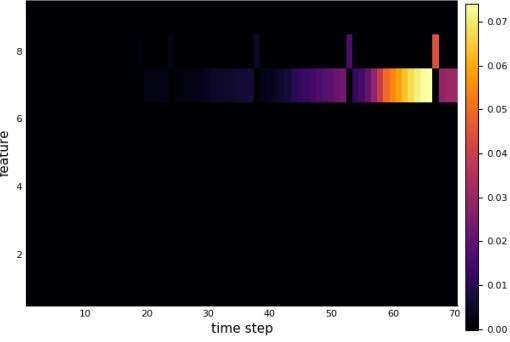

Figure 10: Attributions Result on LSTM Model.

## Discussion

While several explanation algorithms have been successfully used on supervised learning problems, direct application to reward based controls learning is somewhat illusive. A large part of this is due to the time-extended property of control policies. An action taken at a particular time step may seem sub-optimal at that particular time step but has benefits for future time steps. This limits a lot of direct application of supervised learning explanation as these explanations will tend to explain the superficial benefit of the action for the immediate time step and will likely miss the explanations of the future benefits. Our use of grammar-Based decision trees

and temporal modeling attempt to address this issue, but they also lead to another problem: Control policies that need to optimize for future time steps are performing operations that are inherently complex and are difficult to summarize with simple explanations. In our test-domain the explanation algorithms are able to expose a major flaw in the operation of our learned neural network controller. However, it seems unlikely that they would be able to reveal more subtle issues or would be able to scale to more complex learned controllers. In addition the explanations do not seem as convincing or as useful as the explanations the same algorithms provide for their original supervised learning domain.

## Conclusion

Explaining a control algorithm based on machine learning is difficult due to the black-box nature of machine learning algorithms and the time-extended properties of control problems. In this paper we attempt to explain such a controller used on a simple obstacle avoidance problem: a neural network trained using a Monte Carlo algorithm. We do this by applying a number of explainability algorithms to this problem. These algorithms look at the inputs and outputs of the controller and based on these values attempt to explain what the controller is trying to do. The explanation algorithms proved useful in revealing a potential hazard in the controller, where it tries to head towards an obstacle and then turn to avoid it. However beyond this flaw it was difficult to gain deep insights into these explanations.

## Acknowledgments

This work was supported by the ATTRACTOR project within NASA's Convergent Aeronautics Solutions (CAS) program.

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
