# OpenReview forum: "Challenges of Explaining Control"
_icaps-conference.org/ICAPS/2019/Workshop/XAIP — XAIP 2019_

### Official Review · AnonReviewer2 · 2019-05-08
**The paper needs significant improvements**

**Rating:** 2
**Confidence:** 2

**Review:**

This paper treats the problem of providing explanations for a multilayer perceptron which is used as an inverse controller for rover movement in an environment with obstacles. Authors employ several machine learning techniques to provide insights and explanations for this specific problem.

Authors provide several good ideas on how to explain a black-box model. However, there are several flaws in the paper, which can be improved with better analysis of results and defining the hypothesis of the paper. The paper structure and writing style also could be significantly enhanced to convey the right messages. Also, a reader is missing punchlines and clear conclusions and contributions of the paper. Authors treat each ML technique separately, and there is no systematic comparison of the outcome of the methods. Also, the paper is not addressing and discussing the proposed approach concerning state-of-the-art works. (Authors could, e.g. take a look at DARPA survey on XAI).

Some more detailed comments:
- The title does not match the paper content. Authors should be more precise and careful when using some standard terminology such as planning, control, controllers, machine learning algorithms.
- The introduction is too high level and authors address ML techniques too general. This story could be shortened and some discussion and comparison to the related work on explainability of NN and black box models, in general, could be added.
- It seems that there are many authors assumptions encoded in the design of explanation algorithms. Also in some cases, the explanation algorithm simplifies the learned function of NN controller. How could these approaches be generalized to other structures of NNs or for a different control problem?
- Another concern, also highlighted by authors, is that most of these explanation algorithms are still hard to interpret especially by non-technical user. A systematic comparison based on some interpretability measure, accuracy as well as informativeness between provided explanations would be useful for a reader.

---

> ### Author Response · Authors · 2019-05-13
> **Response to Review**
>
> Thank you for your helpful review. We want to attend this workshop to discuss tradeoffs between explainability techniques - a number of which had weak results in our domain and we think it would be a great workshop discussion on how to improve the results. These inconclusive results made it somewhat difficult to write a strongly worded paper, though reading the review, it is now clear that we need to make our goals for this paper more explicit. We think we can do this before the workshop and address the reviewers suggestions about improving the title, including related work on explainability and discussion on the assumptions that these algorithms are making.

---

### Official Review · AnonReviewer1 · 2019-05-09
**Too many spinning plates and vagueness, dive into the specifics and nuance rather than just stating "Wow, such nuance."**

**Rating:** 3
**Confidence:** 2

**Review:**

If you state

"Providing explanations of machine learning is a very ac- tive research field. Several approaches have been proposed for standard supervised learning algorithms."  You should cite relevant articles supporting it.

Pg 2, 2nd paragraph:  You introduce the concept of "local" vs "global" explanations without fully defining or discussing them.

I'm a bit confused at the end of the introduction, is the intent of the paper to compare and contrast the variance of explanation generation within the 6 techniques discussed? Or to showcase that explanations are important? Also, in what ways are these algorithms/techniques going to be tested?

There was mention of "intuitive" explanations, is this tested against a group of Users?

Why do you display and show Equation (1) but then define a second one in the same paragraph but *don't* define it as Equation (2) ?

Don't use a Figure Caption to explain stuff, define that in the text. Then use the Caption for a Label. "The 8 sensors of the Rover" or some such.

Also, try to keep figures to the top or bottom of a column, not in the middle (see Figure 3)

Figure 3, the description of Figure 3 and the labels used for the two charts leave me very confused. Additionally, the caption is entirely too long again. SubFigures would have greatly helped readability.

The ending of the paragraph after Figure 3 is confusing to read and would benefit from a re-write. It's very jumbled and feels like its trying to say too many things. Is knowing about a unit test important to the overall reason you want to argue for explanations?

Wait.  Why is the problem statement "In this paper we attempt to..."  Hidden on page 3 in the middle of a paragraph. This should be known to me in the introduction! I care way more about that sentence than having a walk through of the paper structure -- especially when I'm not even sure of the paper's point yet!

This leads me to another thing I've been having issue with on these first 3 pages.  The section headings are too generic, I don't understand the flow.  Everything inside the "test problem" section is like setup and background to eventually test those 6 methods right?  That isn't actually stated anywhere. Additionally, the motivation behind utilizing the "Test Problem" is left for the reader to decipher a bit. Furthermore, if all of this is setup to begin testing things later on -- are these equations really necessary to know about?

First Paragraph, 4th Page, Last Sentence. While I understand the intent of this sentence, it alludes to a longer/deeper conversation that isn't given justice. Avoid the "We didn't do it that way, cause that's hard." Line, allude to future endeavors or discuss the tradeoffs. This sentence seemingly does neither.

Figure 4 needs way more context for me. This isn't intuitive to read at all. Let alone obstacle vs not. Also, *how* does AA Figure 4 confirm Bayesian Rule List explanations?  You just say it does, what's the connection.  Spell it out for me. Please.

Gradient Analysis: You discuss this without discussing it. This is starting to feel hand-wavy in its lack of detail. Figure 5 needs a lot of explanation and set-up. I have no idea what the numbers are for or why blue dots have anything to do with the action "up".

Explanation Template: Equations, "...shows that...." How? Explain this to me.  Why / How?

Fig 6,7,8 I'm not sure these are all needed as figures. You... barely touch on them or discuss them in text? Are you arguing that these figures are sufficiently descriptive enough on their own? For instance, Fig. 8 you specify your own full grammar, but then move to figure 9 2 sentences later.

Conclusion Section: No conversation on follow-up? Ideas to facilitate improvement to one of the plethora of techniques introduced in the paper? An alternative *not* pursued by the paper?  What's the next beat of this research?



Minor Things:
  Please number your sections. Easier to reference.

  Using lists and numbering things without breaking it out into new lines isn't the easiest to read at a glance. Either just list the things without numbers, or separate it out.

  The "overview paragraph" at the end of the introduction starts as a list of things, then moves to a conversation/defense of what's in the paper. It is a bit odd, pick one or the other.

  Why is there a reference for saying your robot senses using 8 sensors. This seems odd.

  Figure 2, if you try to reference parts of a Figure make it sub-figures "Refer to Figure 2a" as compared to "Refer to the Left side of Figure 2". Or just state it as a sentence and not a parenthesized referral.

  I'm not sure the workshop's view on footnotes, the footnote 1 can just as easily be stated within the paragraph it is relevant to.  This would make it easier to read overall as it is already a busy sentence with the citation in the middle of it. Also, the footnote is on the next line on its own which isn't ideal.

  "200 Monte Carlo runs... " Don't start a sentence with a number.

  Why is there no numberings to the equations on the 3rd page if you started numbering the equations on the previous page?

  "Note that the goal sensor.... " is a good sentence to showcase how to redo that footnote from above. (Likewise, that could become Footnote 2)


  "... possible weight values to be on too narrow of interval for the the neural network... "  ??? what?

References: Make Sure All Titles Have Proper Capitalization.

---

> ### Author Response · Authors · 2019-05-13
> **Response to Review**
>
> Thank you for your detailed and helpful review. We want to attend this workshop to discuss tradeoffs between explainability techniques - a number of which had weak results in our domain and we think it would be a great workshop discussion on how to improve the results. These inconclusive results made it somewhat difficult to write a strongly worded paper, though reading all the reviews, it is now clear that we need to make our goals for this paper more explicit. Addressing your comments will help make this a better paper, especially moving the problem statement forward, making the motivation of the paper more explicit, clarifying the descriptions of our algorithms and having a more detailed discussion. Note also that we agree that having numbered sections would be helpful, but the current style file seems to preclude it - we can see if the workshop chairs will allow a different style.

---

### Decision · Program_Chairs · 2019-05-15

**Decision:**

Accept

**Comment:**

While the reviewers view this paper somewhat critically, in the spirit of making the workshop a venue for discussion and feedback we decided to reject only those papers with strong reject votes.

Please address all review criticism as best possible for the final paper version and its presentation at the workshop. Looking forward to discuss your work at the workshop!